# The People's Speech: A Large-Scale Diverse English Speech Recognition Dataset for Commercial Usage

**Daniel Galvez**
NVIDIA

**Greg Diamos**
Landing AI

**Juan Ciro**
Factored

**Juan Felipe Cerón**
Factored

**Keith Achorn**
Intel

**Anjali Gopi**[*]
Oracle

**David Kanter**
MLCommons

**Maximilian Lam**
Harvard University

**Mark Mazumder**
Harvard University

**Vijay Janapa Reddi**
Harvard University

`datasets@mlcommons.org`

## Abstract

The People's Speech is a free-to-download 30,000-hour and growing supervised conversational English speech recognition dataset licensed for academic and commercial usage under CC-BY-SA (with a CC-BY subset). The data is collected via searching the Internet for appropriately licensed audio data with existing transcriptions. We describe our data collection methodology and release our data collection system under the Apache 2.0 license. We show that a model trained on this dataset achieves a 9.98% word error rate on Librispeech's test-clean test set. Finally, we discuss the legal and ethical issues surrounding the creation of a sizable machine learning corpora and plans for continued maintenance of the project under MLCommons's sponsorship.

## 1 Introduction

End-to-end deep learning has made it possible to efficiently build new speech recognition systems that are highly accurate and can generalize across languages, speakers, and environments. However, training high-quality automatic speech recognition (ASR) systems remain challenging. The accuracy of speech recognition systems depends heavily on the quality of their training data: training models to achieve or surpass state of the art requires a large corpus of data from diverse environments and a wide variety of speakers. Additionally, an under-recognized constraint is the importance of proper licensing of the underlying speech training data.

For example, in 2018, Baidu tried to release publicly the dataset used to develop Deep Speech [14] as a resource to accelerate speech recognition research, similar to ImageNet for speech. During the initial development of Baidu's Deep Speech system, crowdsourcing platforms were used to collect 12,000 hours of read speech from 9,600 English speakers and bootstrap the system. Baidu ran into unexpected legal roadblocks. A legal review concluded that the original license agreement with speakers did not permit the public release or unrestricted commercial reuse, even though it covered usage by Baidu. Because it was infeasible to contact the thousands of crowdsourcing workers to ask them to agree to new terms, the data was never released.

In this paper, we present the People's Speech dataset, a 30,000 hour supervised audio dataset of mostly English speech data that is covered by the Creative Commons Attribution (CC-BY) and Creative

---

[*]Work done while at MLCommons.

35th Conference on Neural Information Processing Systems (NeurIPS 2021), Sydney, Australia.

| Dataset | Commercial Use | Audio Type | Hours | Background Noise |
|---|---|---|---|---|
| The People's Speech | Yes | Diverse Speech | 30,000 | Yes |
| Earnings21 | Yes | Earnings Calls | 39 | No |
| Librispeech | Yes | Read Speech | 1,000 | No |
| Gigaspeech | No | Diverse Speech | 10,000 | Unknown |
| Common Voice (English) | Yes | Read Speech | 1,700 | No |
| MLS (English) | Yes | Read Speech | 32,000 | No |

Table 1: Summary of The People's Speech dataset versus prior work. We focus on commercial use, speaker diversity, quantity, and incorporating natural background noise conditions into the dataset.

Commons Attributions Share-Alike (CC-BY-SA) licenses (permitting academic and commercial reuse to train and evaluate speech recognition systems). We describe our methodology for collecting existing audio data with transcripts from The Internet Archive (archive.org) and aligning the audio to the transcripts. This pipeline is open source under an Apache 2.0 license. [2] The People's Speech dataset is one of the first large-scale, diverse supervised speech datasets under a license permitting commercial usage. Our work demonstrates that it is feasible to curate large-scale, diverse, open and appropriately licensed speech recognition datasets from resources on the web.

This work is a first step towards creating a speech dataset that covers all speakers and environments (not just English audiobooks [23, 25, 26]) for open use [27]. We find that the Internet Archive [4] includes diverse sources of speech beyond audiobooks, including movies, TV, local news, music, historical documentaries, video game replays, stock footage, twitch streams, sports commentary, business news, lectures, sermons, podcasts, old-time radio, court recordings, health, and law enforcement. Audio content with transcripts on the Internet Archive is 1) abundant, 2) searchable by license type, and 3) diverse, facilitating the creation of a our large-scale open speech dataset.

The key insights and lessons learned from our work are as follows:

- **Licensing issues are important to commercial users of datasets, and these issues are solvable.** CC-BY-SA and CC-BY licensed works permit commercial usage and are easy to detect with software, making it possible to build datasets allowing commercial usage from Internet sources.

- **Creative Commons licensed audio data with transcripts is abundant on the Internet but limited to English.** These data are more diverse than audiobooks and indexed by existing platforms such as archive.org. However, these platforms currently index much more English content than any other language, and it is unclear how much non-English Creative Commons licensed content exists in total. The scarcity of non-English Creative Commons data remains a key issue in creating a large-scale, diverse, open and appropriately licensed speech dataset across multiple languages.

- **Large-scale datasets can be created more efficiently than previously thought**. By leveraging technologies like forced alignment, Apache Spark, and hardware accelerators, we can curate a large-scale dataset with minimal cost. We initially estimated that hand-labeling 100,000 hours of speech would cost $5,000,000. Our forced alignment ran on Google Cloud in three days on 52,5000 hours of input and cost an estimated $3,000.

## 2 Related Work

We briefly contrast our work with several related speech datasets. Table 1 highlights the differences.

**Earnings21** [28] is a 39 hour orthographically transcribed speech dataset of public companies' earning calls created by expert transcriptionists and licensed under a CC-BY-SA license. Earnings21 is intended primarily as a test set for named entity recognition on challenging industry-specific jargon. The People's Speech dataset is meanwhile built via forced alignment of audio to existing transcripts.

---

[2]https://github.com/mlcommons/peoples-speech

**Librispeech** [25] is a CC-BY-licensed 1,000-hour standard large-scale speech dataset in the public domain collected from audiobooks of the Librivox project. The most significant criticism of Librispeech today is its narrow setting: Read audiobooks from a single speaker in a clean environment. Our dataset comprises a variety of settings meanwhile.

**Gigaspeech** [18] is a 10,000 hour English speech dataset. Like The People's Speech, it uses forced alignment of existing audio against transcripts to create training data. However, it does not allow commercial usage because its sources may be copyrighted. The usage of data whose creators have not provided consent also raises ethical challenges. Additionally, it uses YouTube videos as a source, which we avoid for reasons described later.

**Mozilla Common Voice** [15] is a CC0-licensed 10,000-hour public domain corpus of single speaker read speech in a variety of languages created by volunteers. Unlike Common Voice, Our dataset uses existing audio and transcripts from resources on the web, which are licensed appropriately for commercial usage by their creators (rather than soliciting volunteers to curate the dataset [15]). We also acknowledge that The People's Speech does not have meaningful amounts of non-English.

**MLS** [26] is a CC-BY-licensed 50,000 hour speech dataset that is derived from the Librivox audiobooks. MLS is primarily a multilingual speech corpus, whereas our dataset focuses only on the English language but with a more diverse set of sources.

## 3 Dataset Description

In the following section, all metrics are computed on the "sources" of our data, which amount to 52,500 hours of audio with transcripts, split across 76,503 individual files. The process to convert these into usable training data, described in Section 4.2 reduces the total number of hours to just over 30,000 hours. We believe reporting these metrics on the original data is more reflective of the dataset.

### 3.1 Licensing Description

Training machine learning models on public domain work is widely accepted legally. Indeed, several datasets built on public domain work already exist and were examined in Section 2. Public domain work, not having any copyright protections, allows for unencumbered use for almost any purpose.

Our dataset consists of public domain, CC-BY-licensed, and CC-BY-SA-licensed data. CC-BY and CC-BY-SA stand for "Creative Commons Attribution" and "Creative Commons Attribution-ShareAlike" [1]. Both license types allow authors to give content to users to (1) share the work and (2) adapt the work. We interpret the steps to create a machine learning dataset to fit under these allowances. We exclude CC-BY-NC-licensed (Creative Commons Attribution-NonCommercial) works because our dataset is intended for downstream commercial usage. We also exclude CC-BY-ND-licensed (Creative Commons Attribution-NoDerivs) data because we think that the forced alignment stage of building the dataset makes the dataset a "derivative work." Finally, in our data download page, we separate the CC-BY-SA data (3,100 hours) from the rest (27,700 hours) because some users had legal concerns about training on CC-BY-SA data.

To comply with the CC-BY and CC-BY-SA licenses, we must attribute the original creators of the work. This is done by creating a human-readable CSV file in our dataset that lists the author of each CC-BY and CC-BY-SA work. We note that while our entire dataset is licensed as CC-BY-SA because of the viral nature of our CC-BY-SA sources, a commercial user for whom this is not acceptable could straightforwardly filter out the CC-BY-SA sources using this CSV file. Figure 1 illustrates the total number of hours under each license type in the dataset.

### 3.2 Transcript Content Characterization

While our first goal was to create a dataset suitable for commercial usage, we also desired for the speech to be challenging or at least more diverse than existing benchmarks. To this end, we inferred the breakdown of our dataset by language by running the langid software package [24] to get each transcript's language. Note that transcripts are not always in the same language as the audio. In particular, translations are common in subtitles, which cannot be used as labels. Our forced alignment system filters these out. The details of the forced alignment system are described in Section 4.2.

| License Type | Hours |
|---|---|
| US Government | 28034 |
| Public Domain | 11713 |
| CC-BY-SA | 8273 |
| CC-BY | 4496 |

Figure 1: **Hours of audio by license type.** The "US Government" works are in the public domain. We separate the category from "Public Domain" to highlight how large it is relative to others.

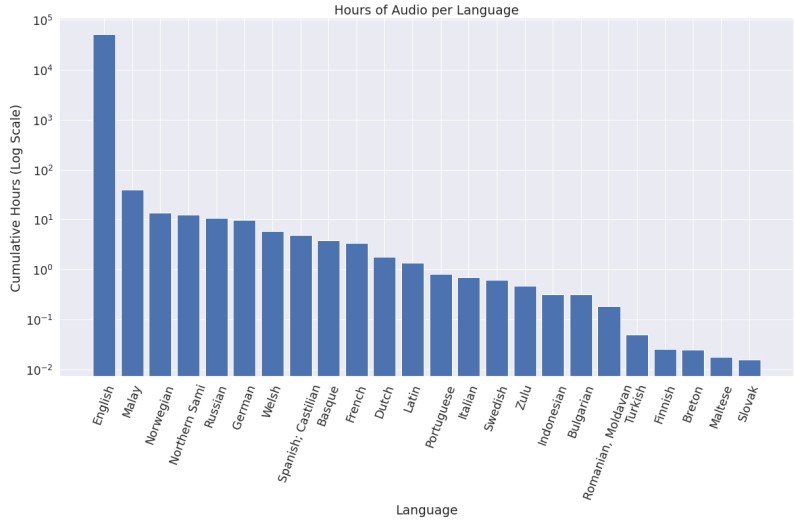

Figure 2: **Hours of Audio per Language.** The majority of our dataset is comprised of English followed by a wide range of other languages.

In total, there are 52,500 hours of audio in our dataset, of which 51,890 hours are English. Figure 2 shows the distribution of hours of audio per language. In addition to English, the dataset includes 23 other languages, albeit with three orders of magnitude fewer hours of audio than English. The following most prominent language in terms of hours of audio is Malay, which has 40 hours. While most languages have fewer hours, small quantities are still useful for tasks such as keyword spotting.

We further characterized the content of the speech data using zero-shot learning to classify the content of our audio transcripts. We used the pre-trained "Bart large MNLI" model to classify the content of the transcriptions into different categories with zero-shot learning [19]. We selected categories that reflected the content we saw in a manually inspected subset of our data: "Government", "Interview", "Health", "Radio", "Lesson", "Finance", "Religion", "Music", "Politics", "Sport", "Sermon", "Entertainment", "Documentary", "Lecture". The two most frequent categories are government and interviews (Figure 5). This matches the observation that around 28,000 hours of our data are licensed as public domain government works (Figure 1). It is important to note that due to the large size of the dataset, the occurrences of the less common categories are not negligible.

We also ran the "Ontonotes fast model" for NER [13] on the speech transcripts and textual metadata to measure the diversity of the dataset in terms of the most discussed topics, etc. The model output 18 different entity tags. The ones we examined were geopolitical entity (GPE) and affiliation (NORP). As Figure 3 illustrates, we find considerable diversity of foreign geographic locations despite the dataset being in American English. While almost all affiliations were American, we were surprised by the many appearances of the word "Tory", referring to the British political party.

### 3.3 Audio Characterization

We analyzed the breakdown by the sampling frequency of our data. Figure 4 shows the cumulative hours of audio across the different sampling frequencies found in the dataset. Almost all data has a sampling rate of at least 16kHz, which means that low-quality data does not need to be upsampled by users of the dataset. Additionally, the high sampling rates suggest the acoustically clean portions of the dataset could be used for speech synthesis in the future.

We also characterized the acoustic backgrounds of our source data. One shortcoming of existing datasets is that they lack the background noise that is representative of a "real world" environment. In practice, background noise is a challenge for ASR systems, so it is important to capture. To examine how "real world" our data was, we used AudioSet's pre-trained YAMNet model [11]. AudioSet is an ontology of various sounds [21]. The pre-trained model performs multi-class classification; specifically, it assigns a probability between 0 and 1 to each class independently, without requiring all probabilities sum to 1 (as you would find in a model using softmax).

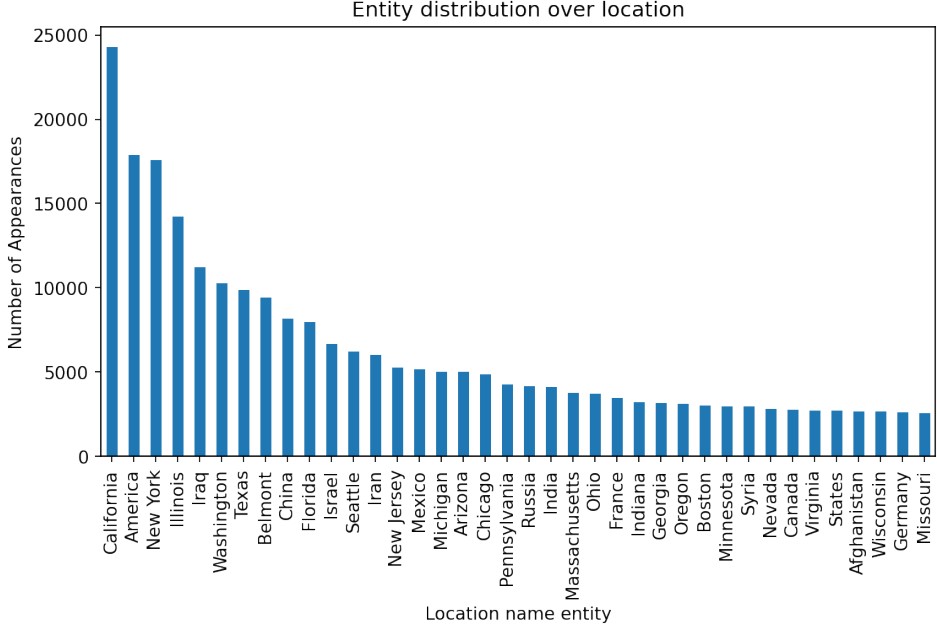

Figure 3: **Number of occurrences of each location in the transcripts.** For clarity, only those places that appear in the data at least 2500 times were taken into account. The "USA" entity was removed because it dwarfed the other locations in number of appearances.

| Sampling Rate | Number of Hours |
|---|---|
| 48 kHz | 17733 |
| 44.1 kHz | 23503 |
| 24 kHz | 1084 |
| 22.05 kHz | 8459 |
| 16 kHz | 988 |
| 12 kHz | 0.5 |
| 11.025 kHz | 40 |
| 8 kHz | 182 |

Figure 4: **Hours of Audio per Sampling Frequency.** Since most speech recognition systems use 8kHz or 16 kHz audio data, our data is suitable for training these models via downsampling.

| Audio Category | Number of Hours |
|---|---|
| Government | 19831 |
| Interview | 10794 |
| Health | 2342 |
| Radio | 2290 |
| Lesson | 2051 |
| Finance | 1757 |
| Religion | 1711 |
| Music | 1690 |
| Politics | 1564 |
| Sport | 1461 |
| Sermon | 1332 |
| Entertainment | 1261 |
| Documentary | 885 |
| Lecture | 595 |

Figure 5: **Hours of Each Category.** Detected by "Bart large MNLI."

Because of time constraints, we randomly sampled 5000 hours of our data rather than operating on the entire 52,500-hour dataset. We ran YAMNet on 15-second chunks of audio. We excluded the "Speech" and "Human Voices" categories from the output because they are not background noise. We estimate each audio category's total number of hours by summing that category's probabilities across all 15-second chunks. We multiply by $52,500/5000$ to estimate the number of hours of each sound in the entire dataset. Results are shown in Figure 6. Metrics like these can be helpful in prioritizing particular data to focus on. For example, while "Basketball bounce" is a relatively rare category, there are approximately 100 hours of it in our data. It probably occurs during a sports game, which may be an acoustically challenging scenario for a speech recognizer in its own right.

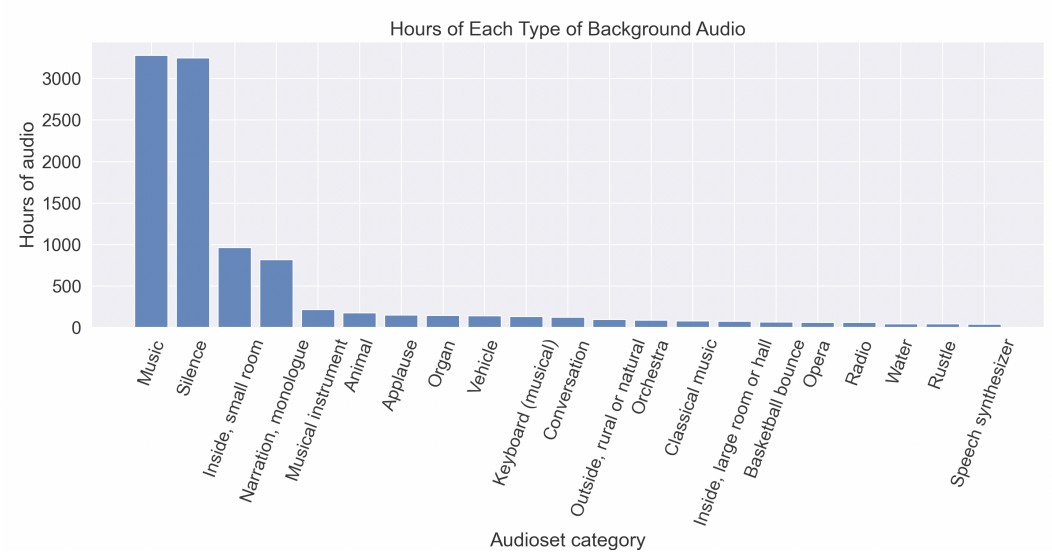

Figure 6: **Hours of Each Type of Background Audio.** Background noise that is representative of "real world" environments is important for datasets that focus on commercial usage.

## 4 Dataset Construction

The People's Speech training dataset was constructed by forced alignment of audio data with transcripts.

### 4.1 Data Acquisition

First of all, we believe it will be useful to the community to understand that it is possible to find commercially-licensed audio content with transcripts via a web crawler. The Creative Commons organization explicitly made the Creative Commons licenses easy for a machine to read, as described in [6]. In particular, HTML pages can annotate particular files as creative commons-licensed with "anchor" tags. For example, this declares the surrounding document is available under the CC-BY 3.0 license: `<a rel="license" href="http://creativecommons.org/licenses/by/3.0/deed.en_US">`. Also, HTML5 videos declare whenever they have subtitles by the appearance of <track> tags underneath the corresponding <video> tag whenever a video has subtitles [3].

In our case, we did not crawl HTML web pages, although we may in the future. Instead, we considered two sources for this work: `vimeo.com` and `archive.org`, both of which index uploaded videos based on whether they are Creative Commons licensed or are otherwise in the public domain [10] [5]. After estimating that `vimeo.com` had 1,500 hours of transcribed audio with commercial-use licenses, we chose to depend on only `archive.org`, which had 52,500 hours of audio with transcripts under commercial-use licenses. The exact code we used to download this audio is available [3]. Notably, we exclude YouTube as a source, whose terms of service claim "You are not allowed to: access, reproduce, download, distribute, transmit, broadcast, display, sell, license, alter, modify or otherwise use any part of the Service or any Content" and "access the Service using any automated means" [12]. Legal counsel advised us that it may not be acceptable to download any content based on these terms of service (even content with a favorable Creative Commons license). In particular, we believe many downstream users would be uncomfortable with such data provenance and reduce the benefits and impact of our work. Recall that our goal is to facilitate broad commercial use.

---

[3] https://github.com/mlcommons/peoples-speech/blob/b7623488dff36d343f8f5a6ead0a5a3a82f723bd/scripts/archive.org/download_items.py

## 4.2 Forced Alignment

Each ground-truth transcript is stored in either SubRip or Web Video Text Tracks format. While both of these formats contain timestamps, we determined via inspection that they were not precise enough. Additionally, we need to remove parts of the transcript that are incorrect. This motivated us to use forced alignment, which is the process of assigning timestamps to words in the transcript.

We forced aligned each audio file in the dataset to its transcript by:

- Transcribing the audio file with a pre-trained deep neural network acoustic model and n-gram language model via Kaldi's GPU-based decoder [16]. The output "hypothesis" transcript contains word-level timestamps.
- Partitioning the hypothesis transcript into chunks of around 15 seconds each, based on the word-level timestamps, which the ground truth transcripts lack.
- Running DSAlign's [2] forced alignment algorithm on each of these 15-second chunks against the ground truth transcript.

The use of DSAlign as our forced aligner is important. We note that our dataset includes the following attributes which make forced alignment difficult:

1. Subtitles to videos are sometimes translations of the spoken language in the audio. In this case, this must be filtered out.
2. Subtitles may not be created by expert transcriptionists, so they may not reflect what was said with 100% fidelity, including disfluencies. They may even be made by automatic speech recognizers. Incorrect portions of the transcript must be removed.
3. Many subtitles are provided by the original video creators and are actually "speaking notes" that diverge from what is said.
4. Sometimes subtitles are descriptive of a video rather than providing a transcript.
5. Audio file duration has a long tail. While the median audio file duration is 0.54 hours, the maximum audio file duration is 13.4 hours. Longer audio files are harder to segment.

That means that we cannot use a straightforward forced-aligner that assumes that the transcript is always correct. We found that DSAlign handled all points well except points (1) and (4). DSAlign assumes there is at least a small overlap between the audio and the ground truth transcript. If DSAlign cannot find any correspondence between hypothesis transcript and ground truth transcript, it runs for hours. We worked around this by running DSAlign on each source audio with a timeout of 200 seconds. 90% of our data went through forced-alignment successfully, while the other 10% timed out. We also computed the character error rate (CER) between the ground truth transcript of each aligned portions against the alignment model's transcription for that portion. If the CER was greater than 50%, we discarded that sample. Out of our 52,500 source hours of audio, we successfully aligned 31,400 hours to transcripts, making our final dataset size 31,400 hours.

### 4.2.1 Forced Alignment Verification

We verified the correctness of our forced alignment procedure in two ways. First, we checked that the model we used for forced alignment had similar hypothesis transcripts to the groundtruth transcripts as measured by character error rate (CER), displayed in Figure 7. Note that, because groundtruth transcripts can be incorrect, we are not concerned about the long tail of aligned samples with very high CER; we simply filter them out, as described in Section 4.2. Similarly, we consider CER rather than the more common word error rate (WER) because we expect our alignment model not to be able to predict the several out-of-vocabulary words in the groundtruth alignments. Since a given training sample transcript is only a few words long, the WER metrics is artificially higher than CER.

Secondly, a trained human transcriptionist manually transcribed one hundred aligned training samples, chosen at random, and the word error rate between the ground truth transcripts and the human-provided transcripts were computed. Results are displayed in Table 2. Noting that human WER on spontaneous speech is around 8%, we expect that 34% of our data (about 10,000 hours) has perfect transcripts, and an additional 22% (about 7,000 hours) has high quality transcripts. However, about 44% (about 14,000 hours) have a higher than human word error rate. Manual investigation

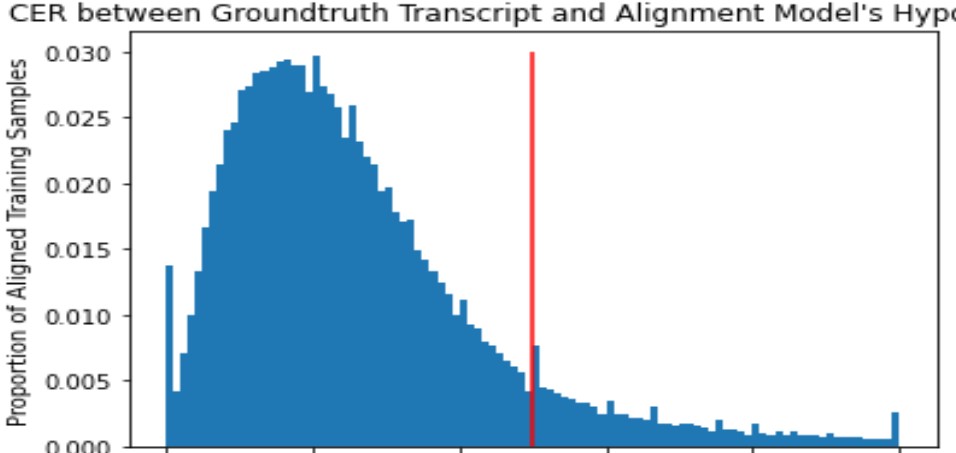

Figure 7: **CER mismatch of aligned samples.** The vertical red line represents our threshold (50%) to consider an alignment correct. Aligned samples with a CER greater than that threshold are removed from the dataset. The area under the curve to the right of the threshold represents 2050 hours, while the area to the left represents 31,400 hours.

revealed that our alignment system sometimes misses the start or end of the transcript corresponding to the audio segment, leading to missing or extra words in the label and a very high number of insertions and deletions for these utterances. We filter out several of these as described in the previous paragraph, but are also working on improving boundary detection in our forced alignment system for the camera-ready paper.

Table 2: Human-vs-ground-truth WER Breakdown

| Sampled utterances | Quality | WER |
|---|---|---|
| 34 | Perfect | 0% |
| 22 | High Quality | 1-8% |
| 44 | Low Quality | 8-35% |

## 5   System Implementation

We include this section to describe some of the challenges with aligning such 52,500 hours of data. We feel that these issues are not discussed well enough in the prior art.

Forced alignment is the most challenging part of the workload. When we started, we used DSAlign's [8] command line interface, which provides a CPU-based pipeline that ran out of the box at 0.5 RTF (real time factor; effectively, it could transcribe 1 hour of audio in 2 hours of wall-clock time), with approximately 99% of the time spent on acoustic model inference, based on profiling with the linux perf tool. Recompiling Tensorflow, the underlying machine learning framework, with the AVX512 instruction set did not provide any meaningful speedup. This motivated the usage of accelerators.

Prior work has shown that once the acoustic model is accelerated on a GPU, roughly 90% of the run time will be spent in external language model decoding on the CPU [16]. Therefore, we were concerned that simply accelerating the acoustic model on a GPU would not give us meaningful overall speed up.

These challenges motivated us to use GPU-based external language model decoder in Kaldi [16], for which the GPU runs both the acoustic model inference and language model decoding, without having to save acoustic model logits to disk to be loaded later by a CPU-based language model decoder. We used only 4 NVIDIA T4 GPUs to align all of our data, with each running at 250x real-time-factor

(i.e., 250 hours of audio could be transcribed in 1 hour of wall-clock time by one GPU). This took only 52.5 hours on our 52,500 hours of source data.

Running DSAlign took approximately another day, putting our total runtime at around three days.

The entry point to our pipeline is at the URL indicated in the footnote. [4]

## 6  Dataset Limitations

In this section, we note some of the limitations in our dataset construction pipeline.

We do not create train, test, and dev splits for our dataset for two reasons. One, we don't have a way to ensure there is no speaker overlap between the splits. Two, we don't have a way to ensure duplicated audio data (there is nothing stopping two separate users from uploading the same audio to archive.org twice) do not end up in separate splits. Were either of these events to happen, the dev and test sets would be artificially easy because of overlap with the train set.

Audiobooks and volunteer-provided audio identify the speaker in metadata. Additionally, there is normally just one speaker. However, for our sources, the speakers are rarely annotated and the speaker set is open rather than closed.

This raises the under-studied importance of automatic speaker identification systems for constructing `test` and `dev` sets. We think locality-sensitive hashing [22] of ivectors [20], xvectors [30], or other text-independent speaker embedding is a promising approach to find speaker overlap in utterances. Since much of our dataset consists of conversational speech, developing an understanding of speaker changes, speaker overlap, and so on, perhaps by running a tool like pyannote [17], is also worthwhile.

Finally, our data is primarily American English. There are a few hypotheses for this situation: (1) `archive.org`, as an American organization, has mostly American uploaders, (2) Creative Commons licensing is not popular outside America, or (3) there is non-English Creative Commons audio available, but it lacks transcripts. If hypothesis (3) is true, we could use weakly supervised training methods on this unlabeled data, like [29]. In particular, all audio data usually has some "context" or "description" in the surrounding HTML page that can be used as weak supervision labels.

## 7  Evaluation

Because we chose not to create a test or dev set for this corpus, we sought to evaluate the performance of an acoustic model trained on the dataset on a test set that lacked overlap with The People's Speech. For this, we chose Librispeech's test and dev splits [25]. We normalized the text of both The People's Speech and Librispeech to use only the lowercase English alphabet, space character and apostrophe. We selected a 20,000 hour subset of The People's Speech that had at most 20% CER between the groundtruth transcript and the pretrained model's output (using the same methodology as described in Section 4.2.1) for training. The Conformer-CTC model architecture [9] was used for training.[5] The beam search decoder used a beamwidth of 1024, language model weight (alpha) of 1.0, word insertion penalty (beta) of 1.0, and Librispeech's pre-trained 3-gram language model with a threshold of 3e-7.[6]

| Model and Word Error Rates | dev-clean | dev-other | test-clean | test-other |
|---|---|---|---|---|
| Conformer CTC Model | 9.93% | 25.53% | 9.98% | 26.91% |

Although the results are worse than the current state of the art on Librispeech, we interpret them to mean that the dataset contains a meaningful signal.

A fundamental principle of machine learning is that train and test distributions should match. The best match for an audiobook dataset like Librispeech would be other read speech datasets like WSJ or MLS. We do not know of a widely used free benchmark test set that is conversational and has

---

[4]https://github.com/mlcommons/peoples-speech/blob/main/galvasr2/align/spark/align_cuda_decoder.py

[5]https://github.com/mlcommons/peoples-speech/blob/1bfaa7d843e0f664e16bbdbc308f7fa40ac7e10c/training/nemo/conformer_ctc_char.yaml was the exact architecture used.

[6]https://www.openslr.org/resources/11/3-gram.pruned.3e-7.arpa.gz

background noise. We expect most users of our dataset to value the dataset for generalizing their speech recognition systems beyond read speech.

## 8 Discussion

MLCommons [7], a non-profit organization related to MLPerf(ormance) benchmarking, intends to sponsor this dataset, as well as future datasets in different domains of machine learning. MLCommons intends to perform the following as its sponsor:

- **Keep the dataset updated.** Datasets must continuously evolve to avoid concept drift. To this end, it is essential for public datasets that are widely used for academic and commercial use to be kept up-to-date. MLCommons makes the commitment to keep The People's Speech updated through periodic revisions. The People's Speech dataset has been developed through a data engineering pipeline that enables us to keep the dataset up-to-date simply by running on new data as it is uploaded to archive.org.

- **Handle legal issues with the dataset.** It is always possible that a particular piece of source data is mis-licensed by its creator. For example, it is possible that a copyrighted song could be playing in the background of an audio recording in the dataset. There needs to be a way for third parties to request removal of copyrighted data from the dataset.

- **Permit easy withdrawal of content.** Allow for creators of content who are unhappy with their works being used as training data to request that it be removed from the dataset. While the Creative Commons licenses that we selected legally permit such use, it is understandable that someone may not have applied such a license with this use case in mind. One concern is that this dataset could enable cloning of the voices of the individuals who appear in the dataset.

- **Fix errors** in the dataset and send versioned updates. Speaking from experience, datasets often have problems that are discovered only after their release. Concretely, we have observed issues like incorrect usage of diacritics in some languages, different text normalization schemes, optical character recognition errors in transcripts of books, or just mistranscribed utterances in several existing speech recognition datasets. Particularly when these errors occur in test sets, it becomes less meaningful to make modeling improvements.

- **Pay for web hosting** of the data. Hosting datasets that are large and commercially useful is a costly enterprise. The People's Speech dataset is 1.12 terabytes in size when compressed in MP3 format. To this end, MLCommons, as the source entity for the dataset, will host the dataset and pay for its hosting fees. The dataset will be available free of charge for its end users.

MLCommons originally developed this dataset because Librispeech, the dataset for its Speech Recognition benchmark, is not representative of the size or domain of datasets used in industry. While representative datasets are available for purchase, the licenses are limited to a single entity. This would require all MLCommons members to purchase a license, which is a challenge for a consortium with over 50 academic and industrial member organizations. Additionally, many legal departments for commercial entities consider benchmarking to be a "commercial" use case, rather than a "research" use case. Accordingly, non-commercial licenses would substantially limit downstream usage. Therefore, MLCommons has a strong interest in continued maintenance of this work.

## 9 Conclusions

We introduce a new supervised speech recognition dataset that sources data from CC-BY-licensed, CC-BY-SA-licensed, and public domain sources. This dataset is large, diverse in scenario, and available for commercial use. We provide our source code to download and force-align the data under a commercial use license as well. Finally, we discuss our forward-looking thoughts on this work.

### Acknowledgments and Disclosure of Funding

We thank Hugo Braun at NVIDIA for assistance with using the Kaldi cuda decoder. Training was done on NVIDIA's internal datacenters. The data download and forced alignment pipeline was run on Google Cloud Platform instances paid for by MLCommons.

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
