# OpenReview forum: "The People’s Speech: A Large-Scale Diverse English Speech Recognition Dataset for Commercial Usage"
_NeurIPS.cc/2021/Track/Datasets_and_Benchmarks/Round1 — NeurIPS 2021 Datasets and Benchmarks Track (Round 1)_

### Official Review · Reviewer_Taxo · 2021-06-27
**A useful dataset that is available for commercial usage**

**Rating:** 7
**Confidence:** 5
**Correctness:** The claims are correct.
**Clarity:** The paper is fairly well written and …

**Strengths:**

1. The dataset is larger than other present corpora and can also be used commercially

2. Used a zero-shot technique to classify the content of the transcripts into several categories, which can be used for other downstream tasks.

3. A large portion of the dataset is in 16kHz, presumably of higher quality, and thus can be used for speech synthesis as well.

4. The authors also tried to estimate different types of noises present in the dataset using a pre-trained YAMNet model. I think this is a good addition as future Speech Enhancement works can use audios from this dataset under different noise labels to show real-world examples and use cases of speech denoising.

5. The limitations are clearly mentioned by the authors.

6. The dataset will be updated regularly with new data, corrections, and also removal of data that might be unsuitable for commercial use.

**Weaknesses:**

1. The authors don't benchmark their dataset on multiple SOTA ASR systems. Currently, the authors train only one ASR and test it on LibriSpeech. I would have liked to see more benchmarks evaluating different ASR approaches.

2. The authors also don't use the full dataset to train their network, which I believe is essential for benchmarking. I hope the authors will do this and report in the final version.

3. I understand it is hard to split the dataset based on unseen identities in the train and test splits. However, such a large dataset should ideally have a common official test set for future research works to compare with. While commercial use cases (which the paper aims largely) may not need such a test set, research works can fairly compare against each other fairly with the addition of a test split.

4. I hoped the supplementary material contained a mini-dataset with the same folder structure, file format, etc., of the final dataset. While I can request the data by contacting the given email id in the supplementary material, it would have been easier this way.

**Additional Feedback:**

I hope the noise labeling using YAMNet can be done for the whole dataset and the labels can be provided while the dataset is released. While the labels themselves can be a bit noisy, I think they will be still useful in multiple scenarios.

**Documentation:**

The dataset has been described in detail. Different features like topics,  sampling rates, regions from where the data was originally collected, and other such information is present in the main draft.

**Ethics:**

The authors have discussed different ethical concerns and have agreed to take down samples that might have been wrongly licensed. The authors will also fix errors in the dataset that will come to their notice with increased usage. While such large datasets always run into risks of questionable language content that might be hurtful to different people, I personally believe the benefits of the dataset outweigh such concerns. The maintainers of the dataset should take down any such content when brought to their notice.

**Relation To Prior Work:**

The authors cite all the major datasets that are available at the moment. I would have also cited: https://pytorch.org/hub/snakers4_silero-models_stt/

Their ASR system works really well and presumably has been trained on an extensive speech corpus.

**Summary And Contributions:**

Summary: The authors propose a large scale speech corpus called "The People's Speech". It consists of 31000+ hours of speech data in multiple languages along with their text transcripts. The dataset is aimed for research in automatic speech recognition and can be accessed publicly. The dataset will be available under CC-BY-SA license allowing commercial usage. The dataset is collected by data already available over the internet with their transcripts. The transcripts and audio are then aligned by the creators.

---

### Official Review · Reviewer_rPth · 2021-06-28
**32k hour ASR dataset: good contributions but inadequate supporting evaluations.**

**Rating:** 5
**Confidence:** 4
**Clarity:** Yes, most details are clear.

**Strengths:**

The biggest strength of the dataset is that it is: (i) large-scale, (ii) freely available for academic/commercial purposes, (iii) the pipeline can be used to scale the dataset further, (iv) there is a promise of maintenance, updating, and error-handling.

**Weaknesses:**

The key weakness of the work is: the quality of the collected dataset is questionable. After training on this large-scale dataset, testing on Librispeech gives a very high WER of 32%. The same architecture if trained only on Librispeech gives a WER of 4%. The authors acknowledge this massive gap and I agree with the authors that there is a meaningful signal from the dataset.

But, this massive gap calls into question if a large portion of the data is noisy, such as  (i) incorrect ground-truth text transcripts, (ii) speech corrupted with background speakers/very high noise, etc. If it is not an issue of quality, and only of domain gap (the collected dataset is conversational as opposed to Librispeech), is it possible to get comparable or better WER by finetuning on Librispeech? At the current version, it is hard to see if the dataset is of “usable” quality. Another measure could have been a mini human evaluation, where a random subset of the data can be judged by the evaluators for correctness.

The other weakness (exacerbated by the above) is that the impact of this dataset is not demonstrated in any way. For example, is it possible to show the cases where a model trained on this new dataset will be superior compared to a model trained on one of the previously existing datasets (CommonVoice, Gigaspeech)? Direct comparisons with previous works are not explored.


**Additional Feedback:**

Please check the Weaknesses and Relation to Prior work section. The paper could also showcase some samples from the dataset.

**Correctness:**

The claims by the authors are correct. The quality of the collected data, however, has not been evaluated correctly.

**Documentation:**

Yes

**Relation To Prior Work:**

While there is a discussion on the differences of this dataset compared to previous datasets, there is no empirical experiments done to support the point of differences of this dataset.

**Summary And Contributions:**

Final rating:
I have read the author's response and the updated PDF. I still believe that the dataset is currently not in a usable state because of the high label noise. Future versions might correct this to some extent, but the current version is not above the acceptance threshold.

The work creates a 31000 hour, large-scale Automatic Speech Recognition (ASR) dataset that is (i) allowed for commercial usage, (ii) in conversational style English versus narrated speech such as LibriSpeech. The dataset will be freely available publicly. The code for data collection will also be released. These are the key contributions.

Evaluating how good this collected data is: The paper trains a recent ASR model (Quartznet) on a subset (7000 hr) of the collected data. This model achieves a WER of ~32% on the Librispeech test-clean set. However, the same architecture trained only on Librispeech has a ~4% WER. The authors argue that the reason for this is the domain gap between their large dataset and Librispeech. They interpret that despite this domain gap, the dataset contains a meaningful signal.

---

### Official Review · Reviewer_n7Ar · 2021-07-01
**Solid work on a new audio to text dataset**

**Rating:** 8
**Confidence:** 3
**Correctness:** Yes.
**Clarity:** Yes.

**Strengths:**

- New dataset of diverse 31,400-hour compared to previous datasets with 10,000-hour
- Broad licensing, both educational and commercial.
- The data is used to successfully train models.
- Ethical considerations are discussed and taken into account.



**Weaknesses:**

- The work is not very original.
- Some details are missing.



**Additional Feedback:**

The paper mentions that “Although the results are worse than the current state of the art on Librispeech” - it would be good to have a reference and value from previous literature.

Regarding the dataset characterization, it would make more sense to use for it the final available sample (only 31,400 hours instead of ~50k).



**Documentation:**

Yes.

**Ethics:**

No.

**Relation To Prior Work:**

Yes.

**Summary And Contributions:**

This paper presents a new dataset of 31,400-hour conversational English recordings. The main advantages are the thorough ethical discussion and the proper license for academic and commercial usage under CC-BY-SA.

The paper provides a proper comparison with previous datasets and clearly motivates the need for a new dataset. Compared to previous datasets the new annotation provides more data diversity in terms of contexts for collection and authors- In addition also a commercial licensing.

The dataset is collected via a web search for appropriately licensed audio data with already existing transcriptions.  The matching between audio and text is constructed by forced alignment of audio with transcripts.

There is a dataset characterization for the initial set of around 50k collected hours. Content characterization is done in terms of language, topic, and named entity recognition. Audio content is described in terms of audio frequencies, and acoustics background of the voice.

The new dataset is validated and a model evaluation is conducted. The paper refers that is not proper to create train, test, and dev splits for the given data. There is no way to ensure no speaker overlap between the splits. Despite being a serious problem, this is present also for the previous datasets (e.g Librispeech). The authors solve it by using data from another dataset as the test set.

Although the results are worse than the current state of the art on Librispeech, the results show that a model trained on this dataset achieves a 32.17% word error rate.

---

### Decision · Program_Chairs · 2021-07-26

**Decision:**

Accept

**Comment:**

This paper presents a dataset of31,400-hour conversational English recodings. One major concern is about the quality of the dataset's labels.
However, the authors have performed some further analysis to justify the quality of the dataset preliminarily. I'm inclined to accept the paper. Authors can perform more quantitative analysis about the dataset's quality in the revised version.